# Left-Behind Children’s Positive and Negative Social Adjustment: A qualitative Study in China

**DOI:** 10.3390/bs13040341

**Published:** 2023-04-19

**Authors:** Wen Liu, Yining Wang, Lingxiang Xia, Weiwei Wang, Yongqiang Li, Ye Liang

**Affiliations:** 1College of Psychology, Liaoning Normal University, Dalian 116029, China; 2School of Psychological and Cognitive Sciences and Beijing Key Laboratory of Behavior and Mental Health, Peking University, Beijing 100871, China; 3School of Psychology, Southwest University, Chongqing 400715, China

**Keywords:** positive social adjustment, negative social adjustment, qualitative, left-behind children

## Abstract

Individual interviews were conducted with a total of 66 participants from five groups between May and November 2020: left-behind children, parents, teachers, principals, and community workers. The left-behind children group included 16 primary and secondary school students aged 10–16. Themes in the interviews’ data were identified based on the Grounded Theory. Left-behind children’s social maladjustment manifested as: (1) depression and loneliness; and (2) poor academic performance. Left-behind children’s positive social adjustment manifested as: (1) using adaptive coping strategies; and (2) life skills and independence. Left-behind children’s social adjustment is a dynamic process and has both positive and negative aspects.

## 1. Introduction

Over decades of urbanization and industrialization, China has encountered mass labor force migration from rural to urban areas. As a result, children are left behind by their migrant parents in rural areas; these children are commonly referred to as “left-behind children”. In this study, the operational definition of left-behind children (LBC) is children who have been left alone in their hometown and have been cared for by people other than their parents for over 12 months [1]. In China, over 68 million children were left behind by at least one of their parents in 2015 [2]. Various studies have shown that, as a group of disadvantaged children, LBC have their own adaptive challenges and problems [3,4]. Compared with children in general, LBC have higher levels of loneliness and depression [5,6] and lower levels of school satisfaction and academic performance [7]. Indeed, both cross-sectional and longitudinal evidence have suggested that LBC exhibit social maladjustment in multiple domains [8,9]. Moreover, some evidence indicates that LBC and the general group do not differ in their school performance and delinquent behavior [10]. However, these recently conducted studies indexed their social adjustment too simply, using several indicators and focusing on their negative adjustment. This is not comprehensive enough to assess social adjustment since it is a multidimensional concept [11,12].

Compared to children who were not been left behind by their parents, left-behind children were found to have more adaptive problems since they suffered from the lack of care and social support [5]. However, the most prominent problems of LBC’s social adjustment remain poorly understood. In addition, a host of empirical evidence substantiates the notion that individuals who undergo adversity could also experience positive adjustment outcomes [13,14], though prior studies rarely focused on positive adjustment in LBC. Therefore, this study aimed to find the salient themes of LBC’s negative and positive social adjustment.

As an important aspect of children’s developmental outcome, social adjustment has been widely explored for many years. With the rapid development of modern society, social adjustment becomes increasingly complex. As individuals enter the phase of childhood, their social environment expands and changes rapidly; every child needs to constantly adjust his/her cognition, emotions, and behavior to adapt to social environments, including the family environment, school environment, peer groups, online communities, etc. [15]. In this process, children constantly interact with families, schools, society, and other proximal and distal environments, coping with challenges in various fields, such as academic achievement, establishing interpersonal relationships, acquiring social norms, and experiencing certain self-experiences and emotional states [16]. Adjustment levels in each domain promotes and compensates the others, forms a certain level of psychological and behavioral adjustment, and, eventually, integrates into a comprehensive social adjustment level, which is closely related to children’s physical and mental health [3]. The formation and the improvement of social adaptability in childhood are necessary prerequisites for children’s positive development in the following stages of life by preparing them to be confronted with a wealth of challenges.

To date, the definition and the structure of social adjustment in psychology are varied due to different research purposes and targeted groups. Firstly, the understanding of the term “social” in social adjustment is varied. Some researchers pay more attention to social interaction and believe that interpersonal interaction is the motivation behind social adjustment in children [17]. According to the objects and scenes of social interaction, John evaluated children’s functioning in school and spare time activities and with peers, siblings, and parents to assess their social adjustment [18]. Some researchers pay more attention to social roles and socialization tasks. They believe that social adjustment is the matching degree of one’s behaviors and his/her social role and that it is motivated by the social requirements from the environment [19]. Zou and his colleagues indexed social adjustment by self, behavioral, environmental, and interpersonal adjustment based on their social developmental tasks, which is a widely recognized comprehensive model of social adjustment in children and adolescents [15]. The establishment of both social interaction and social role are important motivations and indicators of children’s social adjustment. Social interaction is focused more on the person–person relationship, while social role is focused more on broader person–environment relationships, which include person–person relations. Hence, the perspective from social role is more comprehensive.

Secondly, researchers have also created different definitions of “adjustment”. Most studies take the outcomes or results of social adjustment in children as including emotional adaptation, behavioral adaptation, interpersonal adaptation, and academic adaptation [20,21,22]. Additionally, some studies pay more attention to the traits of adaptability or the personal states of children and divide them into psychological advantage, psychological energy, interpersonal adaptability, and psychological resilience [23]. In addition, some researchers examine social adjustment from the perspective of coping and adaption processing [24] and have supposed that there are five aspects of social adjustment: resourcefulness and focus, being physical and fixed, alert processing, systematic processing, and knowing and relating [25]. As a dynamic process, social adjustment is neither linear nor lockstep [26], though there might be several specific fixed characteristics of social adjustment in certain groups, such as people with a certain disease or experience with negative life events. Therefore, it is essential to qualify social adjustment based on its internal states, adoptive strategies, and social outcomes in children.

Thirdly, researchers also have different views on the form of social adjustment. Many researchers have explored negative social adjustment, believing that social maladaptation is the main form of social adjustment [27]. However, other researchers have explored positive social adjustment. Mahoney and Bergman defined social adjustment as the developmental processes by which individuals attain unusually beneficial adjustment patterns given their background and available resources [28]. Clearly, taking both positive and negative adjustment into consideration is more beneficial to empirical work aimed at preventing and intervening in cases of maladaptation, as well as promoting positive adjustment in LBC.

Based on previous studies and arguments, the current study adopted the person–environment perspective and aimed to understand LBCs’ positive and negative social adjustment, including their internal states, adaptive strategies, and social outcomes. In addition, we interviewed LBC, parents, teachers, community workers, and principals to gain a better understanding of LBCs’ social adjustment from multiple perspectives. It is of great theoretical and practical significance to explore left-behind children’s social adjustment more comprehensively as it lays solid foundations for exploring protective and promoting factors, as well as for the development of relative intervention programs.

## 2. Method

### 2.1. Participants

Based on the definition of LBC brought by Lin and Yuan [1] and using the purposive sampling technique [29], a total of 66 participants were selected. Firstly, we targeted one rural area in a northern province and one rural area in a southern province in China; both regions are typical rural areas with a big population of left-behind children with similar economic developmental levels. In each area, participants were recruited from one primary school, one secondary school, and the communities where the schools were located.

There were five participant groups: left-behind children, parents, teachers, principals, and community workers; we chose these groups to provide multiple perspectives on LBCs’ social adjustment at home and school, as well as individually and at the group level. The number of each group of participants interviewed are shown in Table 1. LBC group included 16 primary (grade 4 to grade 6) and secondary school (grade 7 to grade 9) students aged 10–16 (8 girls), with 2–3 students selected from each grade. In each grade, we selected both girls and boys. The parents group included the 12 fathers or mothers of the left-behind children interviewed; their age range was 37–44. The teachers group included 28 class teachers or teaching directors responsible for the left-behind children interviewed. They had ample experience in helping left-behind children; their age range was 30–51. The principal group included 3 principals or deputy principals aged 44–49 for whom caring for left-behind children was a part of their job. The community workers group included 7 chairmen of women’s federations or village secretaries aged 35–64; these individuals had engaged in children and women’s work for over ten years and had focused on the left-behind issue for many years. Ethics approval was granted by our university, as well as the schools and communities/villages through which recruitment occurred; every participant’s consent was obtained before the study. The confidentiality and anonymity of all participants was always maintained; all participants were given pseudonyms.

### 2.2. Interviews

The individual interview method was used to carry out research. This method guaranteed greater individual privacy and ensured that participants expressed more real thoughts, deep attitudes, and personal experiences that fully explored the nature of the questions [30]. A semi-structured interview protocol was undertaken based on the key research questions. Questions were open-ended and has four topics. The first question was “What’s your overall understanding of children’s social adjustment?”. Through this question, the interviewer could clarify the main goal of the study and help participants to understand the meaning of social adjustment in psychology. The second question was “What are the aspects of children’s social adjustment?”. This gave the interviewer the opportunity to help participants understand the nature of children’s social adjustment to identify the important criteria that children must meet to successfully adjust. Moreover, this question could help participants to answer the remaining two questions. The third and fourth questions were “What are the characteristics, strategies, and performance of left-behind children who have great social adjustment level?” and “What are the characteristics, strategies, and performance of left-behind children who are socially maladjusted?”, respectively. Through these questions, we obtained information about left-behind children’s social adjustment in both positive and negative terms. For LBC and parent interviewees groups, they talked about themselves and their children at an individual level, whereas for other groups the content obtained was at the group level.

Before the formal interviews, pre-interviews with five individuals from every group were conducted; we then adapted the language style and the way of raising the questions to suit the needs of each group, especially left-behind children. After every question, the participants were encouraged to give examples related to their daily life to help them fully answer the question.

### 2.3. Data Collection and Analysis

The authors conducted the interviews from May to November 2020. Interviews with children, principals, and some teachers were conducted in the psychological counseling room or empty meeting room of their schools. Interviews with the other teachers, parents, and community workers were conducted by phone and Tencent meetings as required due to the lockdown caused by COVID-19 from May to July 2020. After obtaining the consent of every participant, the entire interviewing process was recorded by two devices in case of device failure. Interview times ranged from 40 to 80 min. The participants received 80 RMB for their time and effort. All recordings were transcribed and rechecked verbatim by the authors directly after the interview.

All the transcripts were managed and further coded by Nvivo12.0 to ensure the consistency of the coding process. The coding of interviews was conducted according to the Grounded Theory [31]. There are three steps of coding: open coding, focused coding, and theoretical coding. In open coding, under the principle of ensuring openness, accuracy, conciseness, and other grounded theories, 10 case texts were named that related to words, sentences, and events to form the original codes. In focused coding, based on all the case text, we identified the most important and frequently occurring original codes that could explain more text and formed the generic codes. In theoretical coding, the relationships between the generic codes were specified and form the integrated theoretical codes. The authors were divided into two groups and conducted coding simultaneously. Weekly intra-group and biweekly inter-group discussions were undertaken.

## 3. Results

This study aimed to understand the positive and negative social adjustments in left-behind children from the perspectives of different groups of people. Hence, the findings from this study are presented in two parts. In Part 1, there are two salient themes related to left-behind children’s negative social adjustment: “depression and loneliness” and “Learning habits”. In Part 2, there are two salient themes related to left-behind children’s positive social adjustment: “adaptive coping strategies using” and “life skills and independence”.

### 3.1. Part 1: Negative Social Adjustment of Left-behind Children

Depression and loneliness: participants described LBCs’ negative emotional states, which particularly manifested as depressed and loneliness (*n* = 30). They indicated that LBC were susceptible to daily negative events, such as peer conflicts or bad academic performance, and sensitive to teachers’ and peer evaluations. The emotional state is an important aspect of the internal state that helps LBC adapt to environmental changes and daily events:

Teacher Hellen: They don’t talk a lot and sometimes they don’t have any reaction when the teacher asks them something in class or in the breaks between the classes. Based on my observation, they always sit in the classroom by themselves rather than join the spare-time activities like others. Once I found that Brain (LBC, 13 years old) came to school without washing his face and rinsing his mouth, maybe because they don’t have parents to take care of these daily issues. This student does not have any other problem, but I just feel that he is not a child of spirit, lively, charming.

Hellen described the characteristics of LBCs’ emotional state by claiming that they were not active and always looked depressed whether they were with teachers or other students. Most participants in the teachers group indicated that they found LBC to look down-spirited or emotionless:

Left-behind child Wendy: I think I feel lonely always, because school is the only place I can talk to others. When I come back home, I cannot communicate with anyone. So, I like going to school.

Left-behind child Yvette: I really don’t know how to talk to my classmates though I really want to. For example, some girls are fans of one band, but I know nothing about that, so I have no idea what they are talking about.

Wendy felt lonely because she cannot find anybody to talk to at home, since she was taken care of by her grandparents and it was hard for her to share the feelings or daily things with them; however, time spent at school could compensate for that. In contrast, Yvette’s loneliness stemmed from school. She did not enjoy being alone at school. She did not know how to fit in and often thought about other students who liked things with which she was unfamiliar. Some LBC had their own best friends; however, they kept their bad emotions to themselves and felt nobody could understand them, as Betty mentioned:

Parent Betty: He (LBC, his son aged 14) is keeping himself in a box. He was crying sadly once on the phone when I mentioned something, but when I asked why, he didn’t say anything and told me that I cannot help him at all.

Interviewer: Did he tell you later?

Parent Betty: No, but he told me before that he thought nobody understand him even his best friend.

Academic performance: most participants mentioned that LBC have difficulties with their learning and most of them had poor academic performance (*n* = 40). The learning habit was the most significant antecedent to poor adjustment (*n* = 27). Teachers and principals acknowledged that it was hard to help them develop good learning habits without their parents’ supervision and education at home. As participants indicated, LBC had a problem with their time management, homework completion, and making study plans, which are all important learning habits. Thus, LBCs’ academic performance was barely satisfactory; this factor was commonly mentioned by teachers and principals (*n* = 19):

Teacher Lance: Terry (one LBC in his class) must wash his clothes by himself and cook sometimes, so I would say it’s hard for a child to consider housework and homework. I would say based on my communication with his father, I don’t think his father tried to make Terry attach importance to study. Other parents have a lot of question about their kids like how do they performance in school or how are their peer relations, but Terry’s dad just asked me did Terry make any mistake or break the rules because his daddy felt it’s in vain to expect too much of Terry.

Most teachers felt LBC in their class had poor academic performance, with the biggest problem being their learning habits. As Lance mentioned, parents’ absence placed a heavy burden on LBC as they had to undertake more housework than children who lived with their parents. Moreover, there were many learning tasks, such as homework, reviewing the subjects, and preparing for quizzes, which require parental supervision. LBC did not have enough time to undertake learning tasks at home due to the housework and did not receive help and education from the family since their grandparents knew little about education; thus, LBC did not develop positive learning habits. LBCs’ parents, on the one hand, felt guilty about leaving their children at home; thus, parents did not want to place too harsh a burden on LBC. Parents also had difficulties educating their children through video calls alone:

Parent Wilda: Her English is not that good, other subjects are also not that good.

Interviewer: what do you think the reasons for that?

Parent Wilda: She didn’t catch up the previous knowledge and gradually it’s hard for her to act as good as others. And I don’t know English at all, so I cannot give any advice to her.

### 3.2. Part 2: Positive Social Adjustment of Left-Behind Children

Adaptive coping strategies using: participants revealed that LBC developed cognitive and behavioral adaptive coping strategies (*n* = 42), which is an important aspect of coping and adaptation processing (Roy, 2009). In total, 21 participants mentioned that LBC effectively coped with their own negative emotions, while 12 participants mentioned that LBC could deal with interpersonal problems in an adaptive way, such as talking to other students to deal with misunderstandings, which is a problem-focused coping strategy. In general, participants indicated that though LBC were undergoing adversity, they still tried to make progress in schoolwork and activities.

Left-behind children Zak: schoolwork is getting harder after getting into middle school. School work takes up more time even for the summer and winter break. It’s very stressful.

Interviewer: So, what did you after you find it’s hard for you.

Left-behind children Zak: I’m optimistic, I think. I just don’t want myself to become tense, so I make a conscious effort to relax, like thinking about something happy or calling my parents to adjust my feeling. My mom always told me I’m different from other kids, I need to take care of myself, so I need to push myself to keep a good state.

Zak described the inner activities that he learned to be strong and deal with negative emotions or daily events. In this context, Zak acquired coping strategies to adjust his emotional state to confront other challenges. He also mentioned that his faith and belief in his mother helped him to persist when he faced difficulties:

Principal Tina: These LBC did great job in our annual school sports meet. Once they were selected to the sports team, they would turn to their coach ask for help about their training. Many teachers said LBC were inclined to be cheery than other time maybe because they found one thing that they were good at. They work very hard since they get some achievement in this field and become more confident.

Tina found that some LBC would rebuild their confidence through other school activities and make efforts to perform it as well as possible. Even though LBC have worse academic performance, they also want to maintain a positive self-concept and fit into the school environment. Many teachers (n = 16) said that LBC did not withdraw themselves from school activities and tried to excel in any activities in which they were able to participate.

Life skills and independence: participants explicitly mentioned that LBC had a great ability to live independently at school snd home (n = 25). They revealed that LBC did not complain about their parents’ absence and acted autonomously when they were at home or school. At home, they could help their grandparents undertake farm work and housework. At school, some teachers (n = 10) indicated that they excelled in empirical tasks, such as experimental science subjects:

Community worker Pamela: It’s very impressive, he (Paul, LBC aged 12) was helping his grandfather doing the farm work, like pouring away the water and pulling up the weeds. It’s unusual now that the kids in our community would help the family to do farm work.

Interviewer: It’s more usual for left-behind children do the farm work in the community, right?

Community worker Pamela: Correct. Other kids have their parents doing that but for left-behind children, they are sensible, they know grandparents are old and not in very good condition, so they are willing to help them with that. And they work effectively and can stand hard work.

Pamela explained that a left-behind children often working in the fields, even though it is already uncommon in this community for children to help with the farm work:

Parent Tina: I always feel surprised that she (Mary, LBC aged 14) can take care of herself and her little brother so well. One example is that she could send her brother to the kindergarten every day and I really think she is a responsible and independent child.

Interviewer: Do you think it’s related to the situation of being left behind?

Parent Tina: I do think so, sometimes I think she is doing our parents duty since we cannot be with them. Also, she could understand our decision to find jobs in another province to give them better life and become more mature in this situation. Me and her father are trying our best to comport her negative emotions at first months of our leaving and make our effort to keep them in touch and feel very delighted she always understand we as a family should fight for life together.

Tan described how Mary approached being left behind. Taking care of family members is a valuable characteristic of mastering life skills. Mary’s attitude and behaviors towards being left behind, as perceived by her mother, changed after her parents helped her to effectively cope with her negative emotion; this change represents a positive adaptive process to the reality of her parents’ absence.

## 4. Discussion

A total of 66 participants were included in five groups: left-behind children, parents, teachers who teach LBC in their class, principals, and community workers with a wealth of experience dealing with LBC issues. They were interviewed about left-behind children’s positive and negative social adjustment, including their emotional states, adaptive strategies, and social outcomes. Qualitative analysis of transcripts indicated that as a disadvantaged group, LBC had both social maladjustment and positive adaptive processes and outcomes in specific domains. In general, the participants in the current study revealed that LBC have four defining characteristics of their social adjustment: (1) depression and loneliness; (2) bad learning habits; (3) adaptive coping strategies using; and (4) life skills and independence. Depression and loneliness were the internal state of social adjustment when preparing to confront the challenges of being left behind. Adaptive coping strategies used the interactive process of social adjustment. Learning habits, life skills, and independence were manifestations of social adjustment outcomes.

After comparing left-behind children and children who live with their parents, LBC were found to have social maladjustment in many domains, such as internalizing problems originating from parents immigrating to other cities [32,33]. However, we would reveal a one-sided view if we were only to focus on the risks to LBCs’ social adjustment. The current study revealed that the pattern of LBCs’ social adjustment was neither positive nor negative as mounting evidence indicated that the disadvantaged children were prone to a broad array of adverse social outcomes; nevertheless, they developed the resilience to respond to adversity and adjust to their environment [34]. Effective parent–child interaction after being left behind also had a compensation effect on LBCs’ well adjustment [35]. In addition, though we found that LBC experienced negative emotional states and poor academic performances, they could still balance the needs of the school, family, and society. Thus, it is clear that social adjustment is multidimensional; individuals with inadaptation in a specific domain would make effort to improve the adaptation level in other domains [15]. Altogether, though LBCs’ maladjustment to their environment stemmed from their parents’ immigration and the risk factors deriving from that, they nonetheless exerted themselves to adapt to family and school life.

In agreement with other studies, being left-behind situation contributed to the development and progression of negative emotional states of social adjustment, especially depression and loneliness [36,37]. Based on the perspectives of LBC themselves and the relevant adult groups defined in this study, LBC exhibited obvious depression and loneliness. Most adult participants described LBCs’ emotional maladjustment as including sadness, inner tension, and an inability to feel; these problems are typical indexes of depression [38]. Additionally, the LBC we interviewed were not talkative and had an unstable emotional state related to their parents’ immigration to other cities. Due to the left-behind situation, LBC were confronted with challenges such as damaged family function, discrimination, and peer victimization, which increased the risk of emotional maladjustment [15,39]. Moreover, previous studies found LBC fared worse than children who lived with one of their parents where emotional problems were concerned [40]. Since both the parents of the LBC we recruited had immigrated, their loneliness and depression were derived from the transcripts distinctly, especially by the teachers.

Concerning academic performance, LBC were found to have worse academic achievement and learning habits as perceived by teachers and principals. One of the most important developmental tasks in childhood is academic achievement, which is also vital to fitting into society [41]. However, in left-behind families, there was often a long-standing dilemma, i.e., though most LBC’s parents held the belief that immigration and education were both important ways of achieving rural-urban mobility, these two strategies caused problems by creating a left-behind family. On one hand, parents’ migration reduced LBCs’ after-school studying time and led to cumulative adaptive problems regarding schoolwork, such as negative learning habits and attitudes [42]. On the other hand, staying in a rural area reduced family income, which might mean that the family could not afford education spending, such as private tutor fees. Hence, it is vital to promote other protective factors to improve LBCs’ learning habits and academic achievements.

Exploring the positive social adjustment in LBC is conducive to finding a way to optimize their overall social adjustment. The findings of this study suggested that LBC used both cognitive and behavioral adaptive strategies to improve their overall social adjustment level, such as engagement with school activity engaging and emotion adjustment, which could also compensate for the inadequate adaptive capabilities in other domains, such as academic performance. Many previous studies considered positive coping as the antecedents to social adjustment [6]; however, in this study social adjustment was seen as a dynamic process that included adaptive coping strategies using [43]. In the context of adversity, LBC used various adaptive coping strategies to adjust their emotions and self-concept to achieve more positive social outcomes, such as peer relationships. Most LBC we interviewed mentioned that they tried to maintain a neutral emotional state since they knew that bad emotions would upset their parents and friends. Moreover, parents and teachers described LBC as thoughtful children who were more considerate and able to adjust their behavior to protect others’ feelings. In addition, the behavioral adaptive strategies used by LBC in this study mainly included engaging with school activities and seeking support; these factors were mentioned by all groups. These results showed that LBC, though in a disadvantaged situation, tried to rebuild their relationships with their peers, teachers, parents, and society in general. It is valuable to know that their adaptive strategies using were salient for different groups of participants, which means they tried to create a virtuous circle in the context of adversity.

Life skills and independence was the most salient theme of LBCs’ positive social adjustment. Kennedy defined life skills as the competencies that an individual needs to sustain and enrich his or her life; these skills are fundamental abilities that ensure social function [30]. For LBC, the need to undertake the responsibilities that their parents would undertake if they had noy been left behind meant that they acquired life skills and independence passively. However, most participants indicated that few LBC would throw a tantrum or complain about their parents’ migrating since their parents left home to increase the family’s income. Besides migration, parental absence could have resulted from personal factors, such as divorce, death, and abandonment. LBC hoped to reunite with their parents and their gratitude motivated them to gain life skills and become more independent [44]. Hence, identifying the leading strengths and weaknesses of LBCs’ social adjustment was the first step to promoting improved quality of life and social adaptation in this disadvantaged group. Additionally, as Mary’s mother Tina mentioned, she effectively copes with Mary’s negative emotions, indicating that how parents cope with children’s negative emotions is crucial to children’s own adaptation. This finding also addressed how LBC’s positive development is to some extent associated with how parents cope with their children’s negative emotions and behaviors [45].

### Limitations and Strengths

While this study provided important insight into LBCs’ negative and positive social adjustment and concerned the internal states, adaptive process, and social outcome of social adjustment, it has several limitations. The LBC and their parents were recruited on the basis that both of their had parents migrated, which may bias the findings since LBC suffered more in this left-behind situation (Wu et al., 2015). Furthermore, no longitudinal data were collected; these data would be necessary to explore the dynamic changes of LBCs’ social adjustment in different domains. For the merits of this study, we interviewed four adult group and LBC themselves to analyze the understanding of LBCs’ social adjustment from different groups’ perspectives. Moreover, it would be of practical value to find the positive aspects of LBCs’ adjustment; data on this subject would enable researchers to probe LBCs’ positive adjustment, as well as the promoting factors driving positive adjustment.

## 5. Conclusions

To identify the salient characteristics of LBCs’ social adjustment, both negative and positive social adjustment were explored. LBCs’ social maladjustment manifested as: (1) depression and loneliness; and (2) poor academic performance, which were the internal state and social outcome of social adjustment, respectively. LBCs’ positive social adjustment manifested as: (1) adaptive coping strategies using; and (2) life skills and independence, which were the adaptive process and social outcome of social adjustment, respectively. Hence, LBCs’ social adjustment as defined in this study is a dynamic process and a comprehensive indicator. This study contributes to this knowledge gap by illuminating how LBC positively adjust their behaviors to suit the disadvantaged environment. The results highlighted the importance of optimizing their social adjustment by considering specific aspects to which they had difficulties adapting.

## Figures and Tables

**Table 1 behavsci-13-00341-t001:** Number of each group of participants interviewed.

	Primary School	Secondary School	Total
Children	7	9	16
Parents	6	6	12
Teachers	12	16	28
Principal	2	1	3
Community workers			7
Total			66

## Data Availability

Anonymized data and details about the preprocessing/analyses are available to colleagues upon request.

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
