# Peer review of "Left-Behind Children’s Positive and Negative Social Adjustment: A qualitative Study in China"

_behavsci, 2023, doi:10.3390/bs13040341_

Round 1
Reviewer 1 Report
This qualitative study conducted individual interviews with 66 participants, including left-behind children, parents, teachers, principals, and community workers in rural China. The authors aimed to gain a more comprehensive understanding of LBC’s social adjustment by focusing on both positive and negative aspects. The authors concluded that, in addition to maladjustment, LBC also demonstrated positive adjustment such as adaptive coping and independence.
The topic is important and interesting, and this report has the potential to be a valuable addition to the literature. However, due to the following issues, I would recommend major revisions before considering publication.
- My major concern is about the interview questions. First, I do not understand the second question: "what aspects of children's social adjustment". More importantly, the third and fourth questions ask about the social adjustment characteristics of LBC who are well-adjusted and maladjusted, respectively, which refer to different individuals, rather than focusing on the coexistence of positive and negative adjustment in the same individuals. The authors should clarify this as it has a huge impact on the interpretation of the results.
- It is unclear to me why academic performance is considered as a part of social adjustment, and why a lack of time for after-school study due to heavier housework is related to “learning habits”.
- The data presentation is too simple and should be elaborated, especially the positive part, which is supposed to be the main contribution of this study. For example, more raw transcripts should be shown (instead of just one example dialog).
- It is unclear how grounded theory contributed to the data analysis of this study. For example, how did the authors arrive at n=42 in line 250 (and many other places)? Some raw coding should be shown and more quantitative analysis should be done.
- It is unclear how the diversity of participants contributed to the study. For example, is there a difference between looking at interviews with LBCs themselves and their parents/teachers?
- The quality of the English writing needs to be greatly improved. Please have a native speaker check for grammatical and usage errors.
Author Response
# Reviewer 1
Thanks for the valuable comments and advice of reviewer, the detailed answers are followed and the revisions were in blue in both of the answers and manuscript.
Comment 1:
My major concern is about the interview questions. First, I do not understand the second question: "what aspects of children's social adjustment". More importantly, the third and fourth questions ask about the social adjustment characteristics of LBC who are well-adjusted and maladjusted, respectively, which refer to different individuals, rather than focusing on the coexistence of positive and negative adjustment in the same individuals. The authors should clarify this as it has a huge impact on the interpretation of the results.
Answer 1:
For the second question of interview, may due to the translation problem, we meant to ask the important domains that children need to make effort to adjust such as academic and interpersonal relationship. That’s the important question to identify main dimensions of children’s adjustment and inspired participants to answer the following questions.
For the third and fourth questions, actually, we indeed focused on the characteristics of LBC’s positive and negative adjustment in both group level and individuals. For LBC and parent interviewees participant groups, they talked about themselves and their kids in individual level, where for other participant groups, they talked about themselves and their kids in LBC group level.
Comment 2:
It is unclear to me why academic performance is considered as a part of social adjustment, and why a lack of time for after-school study due to heavier housework is related to “learning habits”.
Answer 2:
As social adjustment model proposed by Zou et al. (2012), academic is an important aspect of social adjustment since children’s and adaptive abilities and skills would be employed in this domain in China.
For the second concern, heavier housework would be one of the reasons that LBC cannot develop great learning habits since they don’t have enough time to do some review and preview. The revised part is as followed.
As Lance mentioned, Parents’ absence took heavy burden to LBC that they had to undertake more housework than children with their parents at home. Also, there are many learning tasks such as homework, reviewing the subjects, preparing the quizzes, which need parents or elder to supervise. Hence, LBC didn’t have enough time to do the learning tasks at home due to the housework and haven’t got the help and education from family since their grandparents know little about education to develop the great learning habits.
Zou, H., Yu, Y., Zhou, H., & Liu, Y. (2012). Theoretical Model Construction and Confirmation of Middle School Students’ Social Adjustment Assessment. Journal of Beijing Normal University Social Science, (1), 65–72.
Comment 3:
The data presentation is too simple and should be elaborated, especially the positive part, which is supposed to be the main contribution of this study. For example, more raw transcripts should be shown (instead of just one example dialog).
Answer 3:
We added more raw transcripts to elaborat the results about the positive social adjustment of LBC.
Parent Tina: I always feel surprised that she (Mary, LBC aged 14) can take care of herself and her little brother so well. One example is that she could send her brother to the kindergarten every day and I really think she is a responsible and independent child.
Interviewer: Do you think it’s related to the situation of being left behind?
Parent Tina: I do think so, sometimes I think she is doing our parents duty since we cannot be with them. Also, she could understand our decision to find jobs in another province to give them better life and become more mature in this situation. Me and her father are trying our best to comport her negative emotions at first months of our leaving and make our effort to keep them in touch and feel very delighted she always understand we as a family should fight for life together.
Tan described how Mary behaved towards the situation of being left behind. Taking care of the family member is a valuable characteristic of mastering life skills. Besides, Mary’s attitude and behaviors towards being left behind perceived by the mother changed after the parents cope with her negative emotion well, which is an positive adaptive process in the adversity of parents absence.
Comment 4:
It is unclear how grounded theory contributed to the data analysis of this study. For example, how did the authors arrive at n = 42 in line 250 (and many other places)? Some raw coding should be shown and more quantitative analysis should be done.
Answer 4:
We illustrated the three steps of coding in line with the Ground Theory (Charmaz, 2006). 42 out of 66 participants mentioned LBC has one or more adaptive coping strategies like coping with neagtive emotions and interpersonal problems. As reviewer suggested, we added more information about the coding process and details in results.
The coding of interviews was conducted according to the Grounded Theory (Charmaz, 2006). There are three steps of coding: open coding, focused coding, and theoretical coding. In open coding, under the principle of keeping openness, accuracy, conciseness and other grounded theories, 10 case texts were named in terms of words, sentences and events to form the original codes. In focused coding, based on all the case text, we identified the most important and frequently occurring original codes that can explain more text, and formed the generic codes. In theoretical coding, the relationships between the generic codes were specified and form the integrated theoretical codes. The authors were divided into two groups and conducted the coding simultaneously. Weekly intra - and biweekly inter-group discussions were made regularly.
Participants revealed that LBC developed cognitive and behavioral adaptive coping strategies (n = 42), which is an important aspect of coping and adaptation processing (Roy, 2009). In which, 21 participants mentioned LBC cope with their own negative emotion well and 12 participants mentioned LBC could deal with interpersonal problems in adaptive way such as talk to other students to deal with the misunderstands which is problem-focused coping strategy. In general, participants indicated that though LBC were undergoing the adversity, they still tried to make progress in schoolwork and activities.
Comment 5:
It is unclear how the diversity of participants contributed to the study. For example, is there a difference between looking at interviews with LBCs themselves and their parents/teachers?
Answer 5:
The diversity of participants could provide us multiple perspective of LBC’s social adjustment. For example, LBC may cannot provide the information about their negative adaptive behaviors due the social desirability. Besides, parents and teachers perceived LBC’s behaviors differently as well since teachers observed their behaviors at school and could provide more information about it. As reviewer have suggested, we added the reason of recruiting diverse participants.
There are five participants groups: left-behind children, parents, teachers, principals, and community workers to provide the multiple perspective of LBC’s social adjustment at home and school, individually and in group level.
Comment 6:
The quality of the English writing needs to be greatly improved. Please have a native speaker check for grammatical and usage errors.
Answer 6:
Thanks for the suggestion of reviewer, we invited one native speaker to correct our manuscript’s grammar.
Reviewer 2 Report
The authors present an original work with a well-founded theoretical structure and a methodological correspondence adequate to the former. A substantial part of the references are more than ten years old, probably because there have not been significant advances in the field or because they are fundamental works that should be taken up again, in any case the authors should point out the reason for the use of these references.
The ethical considerations of the work, which was submitted to a committee of the authors' university of affiliation, were adequately evaluated.
The limitations and scope of the present investigation are acknowledged, and some suggestions for improvement of the study are presented.
Author Response
# Reviewer 2
Thanks for the valuable comments and advice of reviewer, the detailed answers are followed and the revisions were in blue in both of the answers and manuscript.
Answer:
As reviewer suggested, we illustrated the reason for choosing some classic studies and added some new references as well.
Zou (2012) and his colleague indexed social adjustment by self, behavioral, environmental, and interpersonal adjustment based on their social developmental tasks, which is widely recognized comprehensive model of social adjustment in children and adolescents.
Additionally, as the mother Tina mentioned, she copes Mary’s negative emotions well at first place which indicated that how parents’ cope with children’s negative emotion is crucial to children’s adaption (Ding et al., 2022). It addressed that LBC’s positive development to some extent associated with how parents cope with the children’s negative emotions and behaviors (Lan, 2023).
The current study revealed that the pattern of LBC’s social adjustment is not good-or-bad pattern, as mounting evidence indicated that the disadvantaged children were prone to a broad array of adverse social outcomes, yet they got the resilience from the adversity and adjust the environment positively (Gilligan, 2010). Besides, effective parent-child interaction after being left behind has the compensation effect on LBC’s well adjustment (Khalid et al., 2022).
Additionally, as the mother Tina mentioned, she copes Mary’s negative emotions well at first place which indicated that how parents’ cope with children’s negative emotion is crucial to children’s adaption (Ding et al., 2022). It addressed that LBC’s positive development to some extent associated with how parents cope with the children’s negative emotions and behaviors (Lan, 2023).
Reviewer 3 Report
It was a great pleasure to read the exploratory study, which illustrates the social adjustments of LBC from the word of mouth of LBC, parents, teachers, principals, and community members who potentially provide tangible evidence. Although the paper has merit, substantial issues need to be adjusted.
Introduction: It is proper to introduce and define LBC from an international and Chinese context. What does LBC mean? What is the reason in China children are left behind in the village? These questions enlighten international readers to understand the context.
The introduction section looks long since it hasn't been organized well. The authors/s need to reorganize this section by creating a sub-section in line with the aim of the study.
Gao, C., Tadesse, E., & Khalid, S. (2022). Word of mouth from left-behind children in rural China: Exploring their psychological, academic and physical well-being during COVID-19. Child Indicators Research, 15(5), 1719-1740.
Lan, X. Plight or light? Elucidating parenting styles' main and interacting effects and BIS/BAS profiles on left-behind youth's self-esteem. Curr Psychol (2023). https://doi.org/10.1007/s12144-023-04334-5
Methodology: The researcher/s needs to inform the research design employed in this study.
The author/s stated that purposive sampling was administered to recruit 66 participants, but what were the criteria?
The author/s needs to explain how the validity and trustworthiness of the data.
Khalid, S., Tadesse, E., Lianyu, C., & Gao, C. (2022). Do Migrant Parents' Income or Relationships With Their Left-Behind Children Compensate for Their Physical Absence? Journal of Family Issues, 0192513X221113853.
Author Response
# Reviewer 3
Thanks for the valuable comments and advice of reviewer, the detailed answers are followed and the revisions were in blue in both of the answers and manuscript.
Comment 1:
Introduction: It is proper to introduce and define LBC from an international and Chinese context. What does LBC mean? What is the reason in China children are left behind in the village? These questions enlighten international readers to understand the context.
Answer 1:
We added the definition of LBC of this study at the beginning of Introduction. We using this creitera to select our participants as well, so we added this content in the method part too.
In this study, the operational definition of left-behind children’s (LBC) is children who have been left alone in their hometown and were cared for by people other than their parents for over 12 months (Lin & Yuan, 2007).
Based on the definition of LBC brought by Lin and Yuan (2007) and using purposive sampling technique, A total of 66 participants were selected. First, we targeted one rural area in a north province and one rural area in a south province in China, which are two typical rural areas with big population of left-behind children with similar economic developmental level. Then, in each area, participants were recruited from one primary school, one secondary school, and two communities where the schools located.
There are five participants groups: left-behind children, parents, teachers, principals, and community workers to provide the multiple perspective of LBC’s social adjustment at home and school, individually and in group level. Number of each group of participants interviewed were shown in Table 1. LBC group included 16 primary (grade 4 to grade 6) and secondary school (grade 7 to grade 9) students aged 10–16 (8 girls) and 2-3 students in each grade. In each grade, we have both girl and boy.
Comment 2: The introduction section looks long since it hasn't been organized well. The authors/s need to reorganize this section by creating a sub-section in line with the aim of the study.
Answer 2:
As reviewer suggested, we added a Separate paragraph to illustrate the aim of the current study.
Based on previous studies and arguments, the current study adopted person-environment perspective and aimed to derive LBC’s both positive and negative social adjustment including their internal state, adaptive strategies, and social outcome. Besides, we would interview LBC themselves, their parents, teachers, community workers, and principals to get better understanding of LBC’s social adjustment in multiple perspectives. It is of great theoretical and practical significance to explore left-behind children's social adjustment more comprehensively to lay a solid foundation of the exploring of the protective and promoting factors as well as the developing of relative intervention programs.
Comment 3:
Methodology: The researcher/s needs to inform the research design employed in this study. The author/s stated that purposive sampling was administered to recruit 66 participants, but what were the criteria?
Answer 3:
In this study, we used definition of LBC as indicated previously and the purposive sampling technique recommended by Oliver (2014) that is a non-random way of ensuring that categories of cases within a sampling universe are represented in the final sample of a project. Finally, we targeted four schools and two communities where the phenomenon of left-behind children is common and typical. In detail, first, we targeted one rural area in a north province and one rural area in a south province in China, which are two typical rural areas with big population of left-behind children with similar economic developmental level. Then, in each area, participants were recruited from one primary school, one secondary school, and two communities where the schools located. Thus, this process is in line with the interview-based quantitative research and get representative sample. As reviewer recommend, we added reference and more information about the sampling process.
Based on the definition of LBC brought by Lin and Yuan (2007) and using purposive sampling technique (Oliver, 2014), A total of 66 participants were selected. First, we targeted one rural area in a north province and one rural area in a south province in China, which are two typical rural areas with big population of left-behind children with similar economic developmental level. Then, in each area, participants were recruited from one primary school, one secondary school, and two communities where the schools located.
There are five participants groups: left-behind children, parents, teachers, principals, and community workers to provide the multiple perspective of LBC’s social adjustment at home and school, individually and in group level. Number of each group of participants interviewed were shown in Table 1. LBC group included 16 primary (grade 4 to grade 6) and secondary school (grade 7 to grade 9) students aged 10–16 (8 girls) and 2-3 students in each grade. In each grade, we have both girl and boy.
Oliver C. R. (2014) Sampling in Interview-Based Qualitative Research: A Theoretical and Practical Guide, Qualitative Research in Psychology, 11(1), 25-41.
Comment 4:
The author/s needs to explain how the validity and trustworthiness of the data.
Answer 4:
As reviewer suggested, we added more information about the coding process and details in results. We illustrated the three steps of coding in line with the Ground Theory (Charmaz, 2006). 42 out of 66 participants mentioned LBC has one or more adaptive coping strategies like coping with neagtive emotions and interpersonal problems.
The coding of interviews was conducted according to the Grounded Theory (Charmaz, 2006). There are three steps of coding: open coding, focused coding, and theoretical coding. In open coding, under the principle of keeping openness, accuracy, conciseness and other grounded theories, 10 case texts were named in terms of words, sentences and events to form the original codes. In focused coding, based on all the case text, we identified the most important and frequently occurring original codes that can explain more text, and formed the generic codes. In theoretical coding, the relationships between the generic codes were specified and form the integrated theoretical codes. The authors were divided into two groups and conducted the coding simultaneously. Weekly intra - and biweekly inter-group discussions were made regularly.
Participants revealed that LBC developed cognitive and behavioral adaptive coping strategies (n = 42), which is an important aspect of coping and adaptation processing (Roy, 2009). In which, 21 participants mentioned LBC cope with their own negative emotion well and 12 participants mentioned LBC could deal with interpersonal problems in adaptive way such as talk to other students to deal with the misunderstands which is problem-focused coping strategy. In general, participants indicated that though LBC were undergoing the adversity, they still tried to make progress in schoolwork and activities.
Round 2
Reviewer 1 Report
Thank the authors for addressing my concerns. The manuscript has been improved. I only have two additional comments:
1. Regarding my first concern, please update the main text accordingly to add some clarifications about the questions that were asked.
2. There are still many English usage issues.
To name a few:
Line 31: "It is not integrated to assess their adjustment level". What does this mean?
Line 211: "she unfamiliar with" --> "she is unfamiliar with"
Line 340: "were found had" -> "were found to have"
Line 384: "proved that", "prove" is too strong.
Author Response
Thanks for the valuable advice to improve our manuscript. For these two additional comments, the answers and modified parts are as following and as well as in the manuscript.
Comment 1: Regarding my first concern, please update the main text accordingly to add some clarifications about the questions that were asked.
Answer 1: We checked and updated the manuscript based on all the previous questions. except the revised parts we submitted previously, the content modifided this time is as following.
Line 147-156:
This gave the interviewer the opportunity to help participants construct the structure of children’s social adjustment to identify the important domains that children need to make effort to adjust. Also, this question could facilitate participants to answer the following questions. Third and fourth question is “what’s the characteristics, strategies and performance of left-behind children who have great social adjustment level?” and “what’s the characteristics, strategies and performance of left-behind children who are socially maladjusted?”. By this, we obtained the information about left-behind children’s social adjustment of both positive and negative aspects. For LBC and parent interviewees groups, they talked about themselves and their kids in individual level, where for other groups, the content obtained were in LBC group level.
Comment 2: There are still many English usage issues.
Thank reviewer’s patience and we modified the sentences mentioned and went through the whole manuscript. The details are as followed.
Line 31: It is not comprehensive enough to assess the social adjustment since it’s a multidimensional concept (Liu et al., 2015; Su et al., 2013).
Line 221: "she is unfamiliar with"
Line 34, 338, 368: “were found to have"
Line 382: “this study suggested that”
Line 369: “in particular”
Line 100: “in LBC children”
Line 305: “really effectively”